# Latent Matrix Completion Model

## Abstract

Large amounts of missing data are becoming increasingly ubiquitous in modern high-dimensional datasets. High-rank matrix completion (HRMC) uses the powerful union of subspace (UoS) model to handle these vast amounts of missing data. However, existing HRMC methods often fail when dealing with real data that does not follow the UoS model exactly. Here we propose a new approach: instead of finding a UoS that fits the observed data directly, we will find a UoS in a latent space that can fit a non-linear embedding of the original data. Embeddings of this sort are typically attained with deep architectures. However, the abundance of missing data impedes the training process, as the coordinates of the observed samples rarely overlap. We overcome this difficulty with a novel pseudo-completion layer (in charge of estimating the missing values) followed by an auto-encoder (in charge of finding the embedding) coupled with a self-expressive layer (that clusters data according to a UoS in the latent space). Our design reduces the exponential memory requirements that are typically induced by uneven patterns of missing data. We give exact details of our architecture, model, loss functions, and training strategy. Our experiments on several real datasets show that our method consistently outperforms the state-of-the-art accuracy by more than a staggering 40%.

## 1 Introduction

**Motivation: missing data.** Missing data is a widespread challenge across various fields, including epidemiology, social sciences, finance, clinical research, computer vision and many more Enders (2022); Baraldi & Enders (2010); Beaulieu-Jones et al. (2018); Gelman & Loken (2016); Little & Rubin (2019); Garcia-Laencina et al. (2010). For example, missing data is observed in epidemiology due to participant attrition and incomplete responses during health assessments Beaulieu-Jones et al. (2018), while in social science research, non-response bias in survey-based studies compromises representativeness Gelman & Loken (2016). Clinical trials also face missing data due to participant dropout Little & Rubin (2019). In finance and economic analysis, missing data occurs frequently, particularly in time-series data where gaps may arise due to reporting lags, data collection constraints, or economic events affecting data availability Garcia-Laencina et al. (2010). Additionally, computer vision encounters missing data when input image files are corrupted.

**Prior work and limitations.** Over the years, various methods have been developed to address missing data, but each has limitations Woods et al. (2021). For instance, **(1)** single imputation methods Zhang (2016) can reduce the variability of the data and introduce bias due to the assumption that one value can adequately replace the missing ones. **(2)** Deletion methods Baraldi & Enders (2010); Newman (2014) can lead to significant data loss and potential bias if the missing data are not missing completely at random. **(3)** Model based methods Enders (2022); Ma & Chen (2018) and **(4)** machine learning methods Khosravi et al. (2020) are computationally intensive and rely on the assumed underlying statistical model, which can lead to biased estimates and misleading conclusions. **(5)** Multiple imputation methods Schafer (1999); White et al. (2011) require complex statistical expertise to implement and interpret results correctly. **(6)** Hybrid methods that include combinations of the aforementioned methods can be difficult to analyze and implement, potentially requiring careful tuning to balance the strengths and weaknesses of combined approaches, and some combinations can probably yield incorrect solutions Elhamifar (2016). Moreover, none of these methods can handle the large amounts of missing data that are present in modern datasets and that are information-theoretically feasible Pimentel-Alarcon & Nowak (2016).

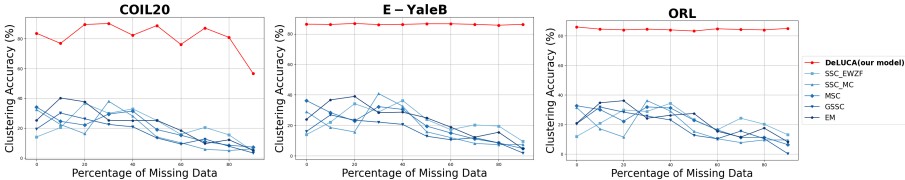

Figure 1: Clustering accuracy for COIL20, Yale B, and ORL dataset respectively. Our architecture outperforms the next best algorithm by a staggering 40% at any interval across all datasets.

Instead of being evenly distributed throughout the feature space, high-dimensional data is sometimes seen to display a low-dimensional structure. Through the use of this structure, observable entries can be completed by interpolating missing values through the inference of the underlying structure. Such completion tasks typically make use of linear subspaces, as exemplified by Low-rank Matrix Completion (LRMC) techniques Candes & Recht (2012).

High-rank Matrix Completion (HRMC) approaches are more effective at approximating modern datasets, which frequently exhibit complexities that exceed the capacity of a single subspace model Eriksson et al. (2012). HRMC accommodates numerous subspaces by extending LRMC with a Union of Subspaces (UoS) framework. This effectively adapts Subspace Clustering (SC) techniques Parsons et al. (2004) to handle incomplete data. The goals of HRMC are as follows: (a) determine the ideal UoS that most accurately depicts the incomplete data matrix $\mathbf{X}^{\mathbf{\Omega}}$; (b) cluster the columns of $\mathbf{X}^{\mathbf{\Omega}}$ in accordance with the determined UoS; and (c) fill in the missing values within $\mathbf{X}^{\mathbf{\Omega}}$. Cluster knowledge would make it easier to apply LRMC to each cluster for data completion and subspace learning; on the other hand, missing value detection would help SC cluster the data and find subspaces. These two goals are intertwined. The main challenge lies in completing these objectives at the same time.

In recent years, there has been a proliferation of High-rank Matrix Completion (HRMC) algorithms due to the widespread adoption of the Union of Subspaces (UoS) model. The algorithms demonstrate a wide range of approaches and levels of performance. Among the notable techniques are nearby methods, which, as Eriksson et al. 2012 describe, use distances between partially seen sites to construct clusters. Van der Velden 2018 also describes naïve methods that replace missing items with zeros or means before clustering using a Subspace Clustering (SC) method. As described by Pimentel et al. 2016a, additional techniques have also been developed, such as GROUSE Balzano et al. (2010), which are used in addition to techniques that combine aspects of ridge and lasso regression. Using unions as second-order algebraic structures in techniques known as "liftings" is another novel strategy Vidal et al. (2005). Furthermore, the incorporation of quantum computing into HRMC represents a breakthrough, as Kazdaghli 2023 demonstrates. Quantum algorithms are utilized for data imputation to boost performance in comparison to conventional techniques. When it comes to neural networks, variational auto-encoders (Kingma & Welling (2013)) and Long Short-Term Memory (LSTM) networks Hochreiter & Schmidhuber (1997) are commonly paired with one another to handle data imputation. In addition, generative models like Denoising Diffusion Probabilistic Models (DDPM) Lugmayr et al. (2022) and Generative Adversarial Networks (GANs) Goodfellow et al. (2014) are being used more and more in matrix completion tasks.

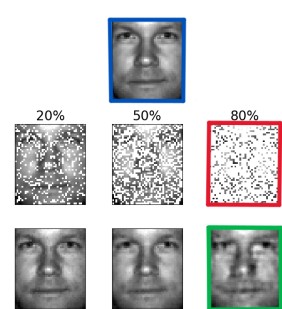

Figure 2: Reconstruction Images for Yale B dataset. The reconstruction capabilities of our DeLUCA model are extraordinary. For example, the image highlighted in green is a reconstruction obtained from the image highlighted in red, which has 80% of their entries missing. Compare to its original, highlighted in blue.

Unfortunately, there are drawbacks to each of these techniques. For instance, naïve approaches face challenges when working with relatively large datasets since the mere act of filling in missing data disrupts the fundamental Union of Subspaces (UoS) structure Van Buuren (2018). On the other hand, neighborhood approaches are not feasible in many situations since they need a super-polynomial number of samples or a significant number of observations to provide adequate overlaps (Eriksson et al., 2012). Increasing the dimensionality of an already high-dimensional space is the goal of lifting methods Vidal et al. (2005). Problems like quantum noise and computational constraints occur in quantum computing, a field within quantum information

theory Kazdaghli et al. (2023). Furthermore, while many approaches produce excellent results on synthetic data, real-world data performance presents difficulties.

**In this paper**, we present *Latent Union Completion* (LUC), a novel and broader completion model. The basic goal is to identify a non-linear structure that encapsulates the observed data and may be represented as a UoS in a latent space. The objective is to use this latent UoS (LUoS) to fill in the missing data. Our proposal involves a deep learning architecture with three main components: (i) an auto-encoder that embeds the data into the latent space; (ii)

| # of Subspaces | 1 | K | K+embedding |
|---|---|---|---|
| Full | PCA | SC | DSC-net |
| Incomplete | LRMC | HRMC | LUC (this paper) |

Figure 3: LUC and related problems.

an auto-completion layer that estimates the missing values in the input $\mathbf{X}^{\Omega}$; and (iii) a self-expressive layer that clusters the embedded data based on a UoS model.In this manner, we can concurrently do both goals of clustering and completing with our architecture, which we refer to as *Deep Latent Union Completion Architecture* (DeLUCA). Above all, our experiments on the COIL20 Nene et al. (1996), Extended Yale B, Lee et al. (2005), and ORL Samaria & Harter (1994) datasets demonstrate that it can achieve extraordinarily high accuracy on real datasets. Figure 1 provides a summary of the findings and allows for comparison with other cutting-edge techniques. This Figure demonstrates how, at every given interval of the proportion of missing data over the whole dataset, our architecture performs $40\%$ better in clustering accuracy than the next best approach. Furthermore highlighting our model's remarkable reconstruction ability is Figure 2. To illustrate an astonishing resemblance to the genuine (original) image, highlighted in blue, the face from the Yale B dataset, indicated in red, was finished from the partial image above, noted in green. The outcomes of the remaining samples are equally striking (see Figures throughout).

Architecture novelty. Another model that has shown success in the related problem of Subspace Clustering (SC) serves as the basis for our design, DeLUCA. Remember that when data is completely observed, SC can be thought of as the particular case of HRMC, where the objective is just to cluster the data based on a UoS. More specifically, we modify the DSC-net architecture Ji et al. (2017) by introducing a unique pseudo-completion layer made up of two partially connected layers as a first component. The main innovation is that the missing elements will be imputed from the normalized entries of the observed data after the data has gone through the pseudo-completion layer. This makes it possible for us to enter data that is incomplete into a clustering network that would not operate otherwise. By including this pseudo-completion layer, our design can now smoothly handle clustering and finishing at the same time.

## 2 LUoS MODEL AND DeLUCA NETWORK

This section contains a detailed presentation of our LUoS model, along with an explanation of the difficulties a deep learning architecture faces in the event of missing data and how we overcome them to create our DeLUCA network. Hereafter, we will use $\mathbf{X}$ to denote a full-data matrix of size $m \times n$, and $\mathbf{X}^{\Omega}$ to denote the incomplete version of $\mathbf{X}$ that is only observed in the entries of $\Omega \subset \{1, \ldots, m\} \times \{1, \ldots, n\}$.

**HRMC** assumes that the rows of $\mathbf{X}$ lie near the union of K subspaces denoted by $\mathcal{U}_1, \ldots, \mathcal{U}_K$. Given $\mathbf{X}^{\Omega}$, the goals of HRMC is (a) to infer the underlying subspaces $\mathcal{U}_1, \ldots, \mathcal{U}_K$, (b) to cluster the columns of $\mathbf{X}^{\Omega}$ according to their closest subspace, and (c) to complete the missing values in $\mathbf{X}^{\Omega}$. LUC. Unfortunately, many datasets do not lie near a UoS. However, any data is more likely to lie near a non-linear structure that can be represented as a UoS in a latent space. That is because UoSs are the special case of this latent model where the embedding is simply the identity map. Hence, we will assume that there exists an embedding $\mathbf{Z} \in \mathbb{R}^{m \times r}$ of $\mathbf{X}$ where the rows of $\mathbf{Z}$ lie near the union of K subspaces denoted by $\mathcal{V}_1, \ldots, \mathcal{V}_K$. We make no assumptions about the number of subspaces or their dimensions. Given $\mathbf{X}^{\Omega}$, the goals of LUC are (a) to find the embedding $\mathbf{Z}$ and infer the latent subspaces $\mathcal{V}_1, \ldots, \mathcal{V}_K$, (b) to cluster the columns of $\mathbf{Z}$ (and by correspondence, the columns of $\mathbf{X}^{\Omega}$) according to their closest latent subspace, and (c) to complete the missing values in $\mathbf{X}^{\Omega}$ according to the inverse embedding and the latent UoS. To achieve these goals, we will use a deep network architecture that is detailed below.

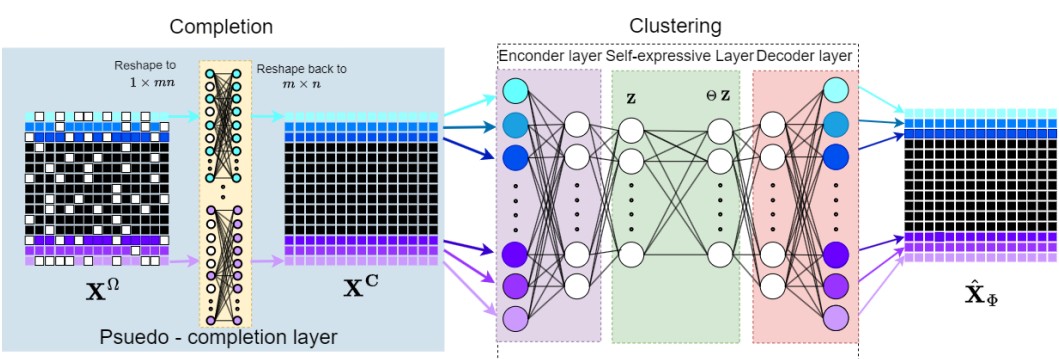

Figure 4: DeLUCA Network.

## 2.1 ARCHITECTURE

Three major parts make up our DeLUCA network (Figure 4): (i) a self-expressive layer, (ii) an auto-encoder, and (iii) a pseudo-completion layer. The pseudo-completion layer, which enables the handling of missing data, is where we mostly contribute. This is by no means a simple task, as each sample contains uneven patterns of observed entries that disturb the data to the point where it can no longer be fed directly into the auto-encoder, as is usually the case with full-data methods such as DSC-net Ji et al. (2017) (for more information, see Section 2.3).

Pseudo-completion layer. This consists of two flattened, partially linked layers. In order to create a completed-data matrix $\mathbf{X}^c$ with placeholder imputations, these layers normalize all of the observed elements in $\mathbf{X}^\Omega$. These normalized values then replace the missing entries. In this manner, the matrix $\mathbf{X}^c$ can be smoothly fed into our architecture's auto-encoder component. See Section 2.3 for more information.

__Auto-encoder.__ The auto-encoder, which performs the task of mapping the pseudo-completed data into an embedding where the self-expressive layer can locate the latent UoS structure that clusters the given data, is the second key element of our architecture. This auto-encoder, as usual, consists of two parts. The encoder is the first component whose goals are to capture important features and patterns in the input data and compress the data into a reduced dimensional space until it generates a compact representation in the latent space. The second part being the decoder, which returns the data to its original space, is the second part of the auto-encoder. Completing the data in accordance with the latent UoS is its goal.

**Self-expressive layer.** The self-expressive layer is our model's final essential element. The auto-encoder contains this layer in the center, in between the encoder and the decoder. Its objective is to use Sparse Subspace Clustering (SSC) to determine the UoS Elhamifar & Vidal (2013). Sparsity in data representations is used in SSC, a technique for exposing UoS structures in high dimensional datasets. The method is to express each sample as a linear combination of the remaining data, so creating a sparse representation of each sample. Subspaces are then revealed by clustering the data according to the coefficients of these combinations.

These three layers achieve their respective tasks through a coupled loss function, detailed below.

## 2.2 LOSS FUNCTION

The design of the loss function required careful consideration in order to accommodate the presence of missing data, and the disparity between the incomplete input and the complete output. Recall that we use $\mathbf{X}^\Omega$ to represent the incomplete version of $\mathbf{X}$ observed on the entries indicated in $\Omega$. Here $\mathbf{X}^\Omega$ is the input of our network. We use $\mathbf{Z}$ to denote the latent representation of $\mathbf{X}$, which corresponds to the output of the encoder and the input to the self-expressive layer, whose weights we represent with the coefficient matrix $\Theta \in \mathbb{R}^{m \times m}$. Notice that besides $\Theta$, the output of our network depends on all the other parameters of the network (weights of the pseudo-completion and auto-encoder), which

we denote as $\mathbf{\Phi}$. To emphasize this dependency, we use $\hat{\mathbf{X}}_{\mathbf{\Phi}}$ to denote the output of our network. Finally, to compare the (complete) output with the (incomplete) input of our network, we define $\hat{\mathbf{X}}_{\mathbf{\Phi}}^{\mathbf{\Omega}}$ as the incomplete version of $\hat{\mathbf{X}}_{\mathbf{\Phi}}$ observed on the entries indicated in $\mathbf{\Omega}$. With this, we are ready to present our loss function:

$$L(\mathbf{\Theta}, \mathbf{\Phi}) \ = \ \frac{1}{2}\|\mathbf{X}^{\mathbf{\Omega}} - \hat{\mathbf{X}}_{\mathbf{\Phi}}^{\mathbf{\Omega}}\|_F^2 \ + \ \frac{\lambda_1}{2}\|\mathbf{Z} - \mathbf{\Theta}\mathbf{Z}\|_F^2 \ + \ \lambda_2\|\mathbf{\Theta}\|_2.$$

**Coefficients and Parameter Tuning.** The first term is minimizing the error between the observed input and their corresponding entries in the output. The second term aims to express each row in $\mathbf{Z}$ as a linear combination of the remaining rows of $\mathbf{Z}$. These coefficients are given by $\mathbf{\Theta}$. The intuition is that $\mathbf{\Theta}$ will reveal the UoS structure because larger coefficients will indicate belonging to the same subspace. The last term aims to regularize $\mathbf{\Theta}$, so that it produces a stable result with minimum norm (as there could be arbitrarily many solutions). We point out that SSC generally uses an $\ell_1$ norm for this last term, in order to favor sparse solutions. Our choice to use the $\ell_2$ norm instead was driven by recent results showing improved performance Ji et al. (2014). In this work it was further concluded that the typical constraint $diag(\mathbf{\Theta}) = \mathbf{0}$ is not a strict requirement when the $\ell_2$ regularizer is used. Finally, $\lambda_1$ and $\lambda_2$ are regularization parameters. These regularization parameters were determined by an iterative refinement approach where parameters were tuned based on their impact on model performance, leading to the identification of optimal values. Our training goal is to find the parameters $\mathbf{\Theta}$ and $\mathbf{\Phi}$ that minimize this loss. We describe our strategy to attain this goal below.

## 2.3 TRAINING STRATEGY

The self-expressive layer requires receiving the entire dataset at once (rather than one sample at a time). This is required because the layer must compute similarities between all samples to learn the patterns in the observed data that will reveal the UoS. Since we are dealing with missing data, it is not possible to pretrain the autoencoder, and hence, we do not pretrain our model. It also does not require initialization of the model with a pre-processed dataset for performing subspace clustering unlike other SC models. It is also to be noted that for training we do not have a set epoch value for termination but rather the termination happens when the learning rate reaches a value of the original learning rate/10. Furthermore, for the loss function, we determine the values of $\mathbf{\Theta}$ and $\mathbf{\Phi}$ by an iterative tuning where multiple configurations were explored, and the final values were selected based on their performance in minimizing the loss.

The additional challenge is that, contrary to what is frequently accomplished by other approaches Yang et al. (2015), the missing entries cannot be naively supplied with zeros or means since this type of imputation introduces bias and distorts the true underlying low-dimensional structure Elhamifar (2016). In a similar vein, we are unable to truncate or sketch (keep only a few characteristics) due to the possibility of bias, information loss, and decreased generalization ability caused by missing data in every column Pimentel-Alarcón et al. (2016b).

It was decided to mask the missing entries so that their absence wouldn't impact the weights assigned to them by the network. But it needed to be done cautiously. The rationale is that every neuron encodes a single complete feature. However, every feature contains a significant number of missing entries under the high missing data regimes that we are interested in. Therefore, masking any neuron with missing values would mask all the neurons, meaning that there would be no active connections between the encoder and the input layer.

**Creation of Pseudo Completion Layer.** We solved the masking problem by flattening $\mathbf{X}^{\mathbf{\Omega}}$ into a $1 \times \ell$ dimensional vector, where $\ell = m \times n$. Now each node contains one entry and all of the missing entries were masked. An additional layer of $1 \times \ell$ dimension was introduced in the model before encoder. For the earlier version of the architecture these layers were fully connected. These layers also served the purpose of preserving the original shape after the masking procedure. The entries in the second layer were then reshaped from $1 \times \ell$ into its original $m \times n$. We termed this obtained matrix as $\mathbf{X}^c$ with $\mathbf{x}_j^c \in \mathbb{R}^m$ representing the feature vectors of $\mathbf{X}^c$. For activation, RELU activation function was implemented in the pseudo-completion layer.

**Refining Pseudo Completion Layer.** But fully connected layers created large requirements of computing resources. To reduce this requirement, connections between initial layers were modified to be partially connected layers. Now each $\mathbf{x}_j^{\mathbf{\Omega}}$ nodes at $m$ intervals in the flattened layer and the

additional layer were interconnected. These partially connected layers together were termed as the pseudo-completion layer.

At the output of the pseudo-completion layer, the data $\mathbf{X}^c$ is now compatible with the auto-encoder. This activates the auto-encoder and it can now perform its function of mapping $\mathbf{X}^c$ into an embedding where the self-expressive layer can find the LUoS structure that clusters the given data. Following the clustering process, the decoder reverses the embedding with the final layer containing $\hat{\mathbf{x}}_{\mathbf{\Phi}_j} \in \mathbb{R}^m$ feature vectors. This is then resolved into an output matrix $\hat{\mathbf{X}}_{\mathbf{\Phi}}$ that comprises of predicted values.

## 3 RELATED WORK

There have been numerous research performed in the field of high rank matrix completion. Long Short-Term Memory (LSTM) Hochreiter & Schmidhuber (1997) is one of the most commonly used methods for this. Further there are multitude of methods which utilize autoencoders for data imputation. Autoencoders are designed for learning efficient representations of data and are used for imputing missing values by training the network to reconstruct the missing values as demonstrated in McCoy et al. (2018). Autoencoders are also used in combination with other methods including PCA, variational autoencoders, denoising autoencoders and, masked autoencoders. One of the major contributions in in image reconstruction field was provided in Zheng et al. (2023) where they had implemented masked autoencoders for preserving intrinsic dimension instead of pursuing reconstruction in the traditional pixel level methods but this method was confined to Deep Neural Networks alone.

MIDAS, a neural network for data imputation was introduced recently which applies the dimensional reduction strategy with an autoencoder to initially corrupt and then reconstruct the image by performing multiple imputations (MI) Lall & Robinson (2022). MIDAS suffers from the usual challenges that MI suffer which include the inability to ignore bias and is also prone to performance loss when subjected to unconventional data structures Lall & Robinson (2022).

Generative Adversarial Networks (GANs) Goodfellow et al. (2014) are also used for matrix completion Zeng et al. (2021); Liu et al. (2020); Yu et al. (2019) but are majorly confined in the image inpainting field. Then Yoon et al. (2018) introduced a new GAIN algorithm that outperformed all the comparative models but also faced all the challenges of mode collapse where the model only generates similar, instead of diverse values. Another research explored was where the structure of a CNN model was inspired by autoencoders and GANs Altay & Velipasalar (2018).

Another new approach is the use of Denoising Diffusion Probabilistic Model (DDPM) for data imputation and image inpainting and has shown to be an alternate paradigm for GANs. The most notable work presented in Lugmayr et al. (2022) in which the RePaint model solely leverages an off-the-shelf unconditionally trained DDPM. Specifically, instead of learning a mask-conditional generative model, they conditioned the generation process by sampling from the given pixels during the reverse diffusion iterations. But, since this is an inpainting method that relies on an unconditional pretrained DDPM. Further a nre diffusion model based method was presented recently for handling missing data Zheng & Charoenphakdee (2023). This method could effectively handle categorical variables and numerical variables simultaneously but the model architecture is still inefficient and requires more optimization.

For subspace clustering, Sparse Subspace Clustering Elhamifar & Vidal (2013) is predominantly implemented. Then Yang et al. (2015) have also implemented SSC while also presenting two new approaches labeled as SSC-MC in which they initially perform matrix completion which they then follow it up by implementing SSC. The second approach was termed SSC-EWZF where they implemented the zero fill method followed by SSC and these two variations of SSC performed best for high dimensional data. They have gotten good results for both synthetic and real data but face the challenge of inaccurate clustering when the subspaces in the dataset are not properly separated and also if the data within them are not well distributed. Another variation of SSC is provided in the Deep Subspace Clustering Network (DSC-net) Ji et al. (2017) which provides performs SSC in combination with neural networks, it solely focuses on the clustering for a complete dataset. It can handle non-linearity which traditional subspace clustering methods may struggle to represent. GANs have also been used specifically for subspace clustering Yu et al. (2022). This paper introduced two new architectures for subspace clustering using GANs but focuses only on clustering.

Generally naive methods offer simplicity in handling missing data. Mean and median imputation, among the simplest approaches, have been employed in diverse fields. For nearest neighbours, more recent contributions in Lee et al. (2012) proposed a novel neighbor-based method for time series clustering Liao (2005), highlighting the adaptability of these techniques to diverse data modalities. Despite these achievements, challenges remain in scalability and parameter sensitivity.

Importantly Lane et al. (2019) adds a valuable layer of insight into our study. This survey paper focuses on completion and clustering accuracy for diverse subspace clustering methods including (1) Group Sparse Subspace Clustering (GSSC), which clusters data points into subspaces while simultaneously promoting sparsity at the group level Pimentel-Alarcón et al. (2016a), and (2) an EM algorithm to deal with missing data in Gaussian mixtures Pimentel et al. (2014) and finally (3) Multiview Subspace Clustering (MSC) Zhang et al. (2018). These methods Lane et al. (2019) offer a benchmark against for our DUC network. It is also to be noted that all the experiments documented in this paper have only been performed on synthetic data. Further, a recent survey paper Cai et al. (2022) that mostly corroborated the results for the methods implemented in Lane et al. (2019).

## 4 EXPERIMENTS

The subsequent sections will describe the baseline models that were used for comparison, how the tests were set up, how the findings were analyzed, and the overall improvements noted in this research. Interestingly, the results achieved for synthetic data are only equivalent to the baseline models, even though our DeLUCA network has shown exceptional performance on real datasets. The synthetic data experiments were carried out with consideration for several factors. These datasets have been diligently constructed to include precisely separated clusters with almost orthogonal subspaces of the same tiny dimensions. The number of subspaces, the even distribution of data across subspaces, the isotropic distribution of data inside each subspace, and the known dimensions were assumed. Furthermore, particular parameters were selected to maximize the performance of the baseline models—basically, customizing the experimental configuration to correspond to their strengths. Even under these idealized conditions, which favored the baseline models, our DeLUCA network produced competitive results. This demonstrates that even while it was tested in a setting designed to bring out the advantages of other approaches, it is flexible and resilient when dealing with a wide range of datasets and circumstances.

**Comparative Baselines.** For the comparative analysis, synthetic and real data were provided to 10 different methods to undergo data completion and data clustering. These methods can be categorized into two types: the first type includes methods exclusively for data completion, while the second type encompasses methods capable of both completion and clustering. The following are examples of completion-only algorithms: (1) SimpleFill: This method uses the most recent non-missing value to fill in any missing values in a dataset. (2) K-Nearest Neighbors: This technique estimates missing values by using the similarity between data points Troyanskaya et al. (2001). (3) Iterative Imputer: Using a sophisticated imputation method, each feature with missing data is modeled as a function of other characteristics, allowing for the iterative prediction of missing values Van Buuren & Groothuis-Oudshoorn (2011). (4) SoftImpute: A penalty term is incorporated while the rank of the finished matrix is minimized to recover missing points Mazumder et al. (2010). (5) MIDAS: To corrupt and recover data, this method uses denoising autoencoders Lall & Robinson (2022).The remaining five models, which are able to do both subspace clustering and reconstruction, belong to the second group. These models, which are covered in the related work section, include SSC-MC, SSC-EWFZ, GSSC, EM, and MSC.

**Synthetic data generation.** A synthetic dataset was generated that is lying near a union of subspaces in the following manner. First, we sampled K d-dimensional subspaces in $\mathbb{R}^n$ uniformly at random by drawing $\mathbf{U}_k \in \mathbb{R}^{d \times n}$ with standard Gaussian entries and orthogonalizing. We then generated data for each subspace, $\mathbf{X}_k \in \mathbb{R}^{m_k \times n}$, as

$$\mathbf{X}_k = \mathbf{V}_k \mathbf{U}_k + \mathbf{E}_k,$$

where each entry in $\mathbf{V}_k \in \mathbb{R}^{m_k \times d}$ is drawn according to $\mathcal{N}(0, 1/d)$, and each entry in $\mathbf{E}_k \in \mathbb{R}^{n \times m_k}$ is drawn according to $\mathcal{N}(0, \sigma^2/d)$. The generated $\mathbf{X}_k$ matrices were then stacked one below the other to form a full-data matrix $\mathbf{X}$ of size n × m. Recall that in our construction each row represents a sample, and each column represents a feature. Finally, recall that each feature vector $\mathbf{x}_j \in \mathbb{R}^m$

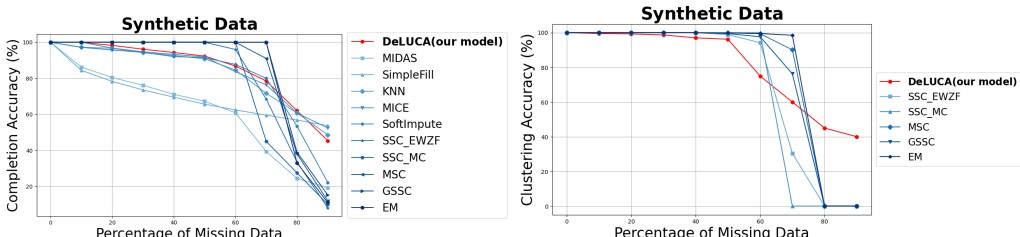

Figure 5: Completion and Clustering Accuracy for Synthetic Data.

contains the $j^{\text{th}}$ feature of all the samples. To create $\boldsymbol{\Omega}$ we sample exactly $\ell > 0$ observed entries uniformly at random.

**Data Parameters.** The synthetic dataset was structured as a $200 \times 50$ (m $\times$ n) matrix and we considered the following parameters: $\text{m}_k = 50$, K $= 4$, d $= 3$, n $= 50$. We included a small amount of noise, $\sigma = 0.01$ where $\text{m}_k$ is the number of samples/subspace, K is the number of subspaces, d is subspace dimension and n is the ambient dimension.

We simulated missing entries by introducing missing values in the synthetic dataset $\mathbf{X}$ dataset with varying proportions, reflecting real-world scenarios where data gaps can emerge due to diverse factors.

**Accuracy metrics.** This synthetic dataset was then fed to the model and the accuracy was plotted against the percentage of missing data in the dataset as shown in Figure 5. For this experiment, we consider two types of accuracy, on our final result $\hat{\mathbf{X}}_{\boldsymbol{\Phi}}$. To measure completion accuracy we take the normalized Frobenius norm of the difference between $\hat{\mathbf{X}}_{\boldsymbol{\Phi}}$ and $\mathbf{X}$, i.e., $\|\hat{\mathbf{X}}_{\boldsymbol{\Phi}} - \mathbf{X}\|_F / \|\mathbf{X}\|$. We additionally measure clustering accuracy which we quantify the proportion of correctly assigned data points to their respective clusters compared to the ground truth. The choice for opting clustering accuracy is due to clustering being an effective method for segmenting images into meaningful regions or for compressing images while preserving essential details. Clustering accuracy is also necessary for preserving quality and integrity of reconstructed features.

As discussed above, it can be observed from Figure 5 that some of the other models for synthetic data slightly outperform DeLUCA in terms of clustering and completion accuracy. See the beginning of Section 4 for a detailed discussion about this.

**Real Data.** Only the results obtained from a real dataset can be used to determine the model's true performance. To generate the missing dataset for this phase of the experiment, several real datasets were employed.

Figure 1 and the other comparative graphs in Figure 6 demonstrate that the baseline models' clustering and completion accuracy may produce remarkable outcomes with synthetic data, but they are unable to replicate the results in real data. On the other hand, our model yielded better results for real data than it did the for synthetic data.

**COIL20** The first real life dataset used in this project is COIL20. This dataset consists of $1440$ $128 \times 128$ grayscale images (20 objects with 72 poses each), with 16,384 features. These images were reshaped to $32 \times 32$ pixels for computation feasibility. The data was fed into the model and it was observed Figure 6 that DeLUCA performed better than all other models in terms of completion accuracy when plotted against the percentage of missing data. It can also be noted from Figure 1 that, the model performs significantly better than all other methods in terms of clustering accuracy and that it outperforms the next best method by more than $40\%$.

From Figure 7 we can observe the image reconstruction performed by DeLUCA for three levels of missing entries starting from $20\%$ to $50\%$ and then finally at $80\%$, in a set of 2 random images sampled from the COIL20 dataset.

**Extended Yale B.** We then used the Extended Yale B which contains $2414$ images of $38$ human subjects with $64$ images per person, where all the images are manually aligned, cropped, and then re-sized to $192 \times 168$ images. From this, we used only 20 human subjects with a total of 1280 images

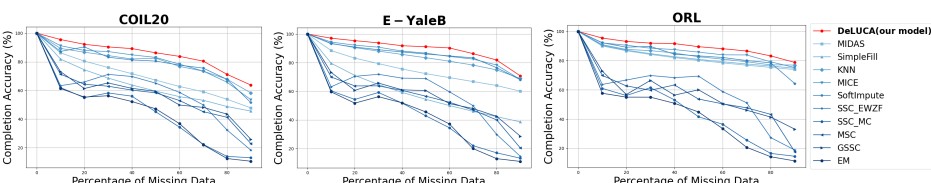

Figure 6: Completion Accuracies for COIL20, Extended Yale B, and ORL Dataset.

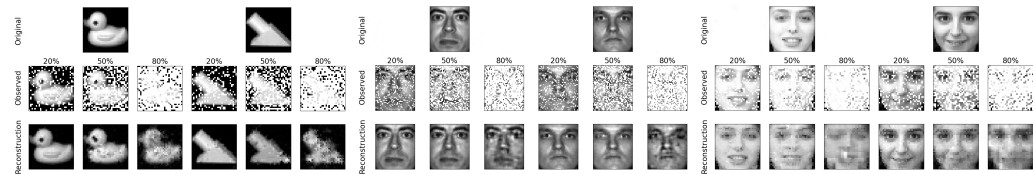

Figure 7: Reconstruction results for different missing data percentages for COIL20, E-YaleB and ORL Datasets.

which were reshaped to $48 \times 42$ pixels. Similar to COIL20, it is again observed from Figure 6 that the completion accuracy of DeLUCA outperforms uniformly over all the other methods for any amount of missing data in the dataset. Again Figure 1 can be referred to see that for even Extended Yale B, our DeLUCA network clearly outperforms the remaining models in terms of clustering accuracy. Also worth noting that for this dataset, even at $80\%$ missing entries, the model performs at a clustering accuracy of more than $80\%$.

Figure 7 shows reconstruction tests performed we used 2 sets of images randomly selected from the Extended Yale B dataset and we imputed the 3 levels of missing entries at $20\%$, $50\%$, and $80\%$ of the dataset.

**ORL Dataset.** And finally, we used the ORL Database of Faces that contains 400 images from 40 distinct subjects. The size of each image is $92 \times 112$ pixels which was reshaped to $32 \times 32$ pixels, with 256 grey levels per pixel. Again as seen in Figure 6, our DeLUCA network provides the best completion accuracy among the state of the art models. The clustering accuracy for this dataset is also similar to what was attained for Extended Yale B as shown in Figure 1.

Figure 7 depicts similar tests which were performed on COIL20 and Extended Yale B, where we randomly selected 2 sets of images each of them for various levels of missing entries were reconstructed.

## 5 CONCLUSION

In this paper we introduced our novel DeLUCA architecture which finds a UoS in a latent space that can fit a non-linear embedding of the original data and performs sparse subspace clustering. We have presented a novel Pseudo-completion layer designed to effectively handle missing data, complemented by an autoencoder that uses the inherent self-expressiveness of latent subspaces for both clustering and data reconstruction tasks. Throughout our research, we encountered significant challenges associated with missing data handling and observed that our model's performance is highly contingent upon the selection of hyperparameters, spanning network architecture and, learning rates. And impressively our experimental findings showcase remarkable enhancements in data reconstruction, surpassing existing models by margins ranging from 5% to 60%, while achieving substantial improvements in clustering performance, with enhancements ranging from 40% to 80%.

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
