# OpenReview forum: "Latent Matrix Completion Model"
_ICLR.cc/2025/Conference — Submitted to ICLR 2025_

### Official Review · Reviewer_q7Cu · 2024-10-24

**Soundness:** 2
**Presentation:** 1
**Contribution:** 2
**Rating:** 3
**Confidence:** 3

**Summary:**

This paper studies missing matrix completion and clustering problems. It improves previous high-rank matrix completion (HRMC) by using non-linear embeddings of an auto-encoder (AE). Besides, the authors propose a pseudo-completion layer, to complete the missing data before being fed into the AE. Experiments are conducted on three datasets, COIL20, Extended Yale B, ORL dataset.

**Strengths:**

Extending the HRMC to non-linear structures seems reasonable.

**Weaknesses:**

1. The presentation is not good and the problem is not clearly introduced. From the abstract, the main problem seems to be learning non-linear embeddings and dealing with the highly sparse observation. These points are not emphasized in the introduction. Moreover, in the paragraph between Line 78-100, the authors list some references like DDPM, GAN, which seem to have no relation with this work. I am still confused after reading the introduction.

2. In abstract, what does exponential memory mean? I do not find further discussion in the main paper.

3. The authors highlighted many keywords at beginnings of paragraphs. While this is okay, the format is not clearly checked and some structures are confusing. For example, some key words are not highlighted as they should be. Moreover, in Page 8 and 9, I think COIL20, Extended Yale B and ORL dataset should not be at the same level with Real Data. The overall readability can be improved.

4. The pseudo-completion layer seems a central contribution. However, I am not convinced about its advantages. Actually, I think there are many similar works regarding VAE with missing data and the authors did not compare them. For example, Mattei & Frellsen (2019) show that zero-imputation works well for VAE. Especially, what is the advantage of reshaping the data into vectors? I am afraid this would significantly increase the computing cost and may fail for high-dimensional data.

5. Experiments are conducted on some  simple datasets. Moreover, the uniformly random missing is also unrealistic in practice.

6. The baselines are not strong and not new. For example, for completion tasks on these datasets, tensor decomposition methods perform quite well and are also very related to this work. I am not familiar with the clustering results, but I am not sure if the improvement is as significant as claimed. In some references in the paper, I can find those models also achieve very small error, for example, Ji, et al. (2017).

7. In Figure 6, the term completion accuracy may be confusing, since the metric is the error norm rather than accuracy.

Typo: psuedo in figure 4.

Mattei, P. & Frellsen, J. (2019). MIWAE: Deep Generative Modelling and Imputation of Incomplete Data Sets. In ICML.

**Questions:**

See Weakness

---

### Official Review · Reviewer_GPui · 2024-10-30

**Soundness:** 2
**Presentation:** 2
**Contribution:** 1
**Rating:** 3
**Confidence:** 4

**Summary:**

This paper presents a new method for non linear high rank matrix completion: the model assumes that each incomplete observation (which corresponds to a column in the matrix) lies in one of several low rank subspaces inside some latent embedding space. The architecture begins with a “pseudo-completion layer”, which is the most/only novel element in the method, which appears to be two trainable fully connected layers . The rest of the model is as in [1]: an auto encoder with a self expressive layer in the middle to perform the clustering (though the authors of this paper remove the typical diagonal constraint $\text{diag}(\Theta)=0$. In typical applications, the samples (columns of the matrix) are images with missing pixels. The latent cluster structure comes from the classes of the underlying computer vision problem. The model is evaluated against many baselines both on a synthetic dataset with latent cluster structure and on three real life datasets: COIL20, Extended Yale B and ORL. In all of these cases, the authors examine how fast the clustering accuracy decreases when the random missingness increases  (and separately, the reconstruction accuracy in L2 norm) and compare it to the baselines. The experiments on synthetic data are inconclusive, with the method being outperformed by some baselines in some cases. The experiments on the chosen real life datasets give “staggering” (cf. line 25) results with a performance increase of 40 percent over the chosen baselines.





**References**



[1] Deep Subspace Clustering Networks. Pan Ji, Tong Zhang, Hongdong Li, Mathieu Salzmann, Ian Reid. NeurIPS 2017.
[2] Sparse Subspace Clustering with Missing Entries. Congyuan Yang, Daniel Robinson, Rene Vidal. ICML 2015
[3] R Mazumder, T Hastie, R Tibshirani, Spectral regularization algorithms for learning large incomplete matrices. JMLR 2010.

[4] Autoencoders with Intrinsic Dimension Constraints for Learning Low Dimensional Image Representation. Jianzhang Zheng, Hao Shen, Jian Yang, Xuan Tang, Mingsong Chen, Hui Yu, Jielong Guo, Xian Wei. ArXiv 2023.

[5] Correntropy-Induced L2 Graph for Robust Subspace Clustering. C Lu, J Tang, M Lin, L Lin, S Yan, Z Lin. ICCV 2013

**Strengths:**

1. The method is simple and could be effective.
2. Figure 4 is quite nice.
3. The results on synthetic data are not good, and the authors admit it, which shows some honesty.
4. Compared to linear methods, the proposed deep auto encoder method performs very well. Independently of whether it would be truly competitive if there were fair baselines, it must be said the results are good in an absolute sense.

**Weaknesses:**

1. Originality and clarity: to the best of my knowledge, **the method** consists in **adding two fully connected layers** at the beginning of the architecture **in [1]** (the authors admit this is the main novelty of the model in line 182: The pseudo-completion layer….. is where we mostly contribute“ . I also read [1], which appears to be exactly the same model apart from the pseudocompletion layer. Thus, the **novelty is extremely limited**. Note that the architectural choice **isn’t properly explained** either: the only description of this key component of the model is “consists of two flattened, partially linked layers... [which] normalize all the observed elements… to create a completed matrix with placeholder imputations…. these normalised values then replace the missing entries”,  see lines 186 to 188). This is not enough information to explain the relevant mathematical operation. Lines 260 to 267 are supposed to add more details to this mysterious description but they don’t add many details apart from the fact that the images are flattened. There is even a sentence which says “For the earlier version of the architecture these layers were fully connected”. This seems more like a  comment left on overleaf for a collaborator than something meant to be in the paper. The authors also didn’t explain what the layers actually are, if they are not fully connected.
2. The **baselines are cherrypicked** and the significance hugely exaggerated. The datasets are small despite the fact that other papers have achieved even more impressive similar results on far bigger datasets such as ImageNet [4].  I don’t usually say this but this is flagrantly true in this case. The baselines seem to include mostly linear methods from subspace clustering [3,2], but don’t seem to include auto encoders. The paper is written in a way that appears to claim that this method is the only one that can achieve the task of simultaneously classifying and reconstructing images, but this is not true.  Here are some important and obvious baselines or experiments to run:
    1. [4] is a framework paper that includes very impressive results on ImageNet for the same task, and it is not included in any of the baselines to the best of my knowledge (I see no -IDC suffix in any of the baselines).
    2. It  would be trivial and very natural as a point of comparison to train [1] (which is almost exactly the same model as this paper) without adding a pseudo-completion layer and to compare this to the performance of the proposed model, but it is not done.
    3. Note also that [5] studies a similar model with even more different missingness patterns for the missing pixels. Not only is this baseline not included.
    4. Similarly to the above points, for an experimental paper like this one, adding more experiments, including studying different missingness regimes would be greatly needed.
    5. Experiments on bigger datasets such as ImageNet are needed.


In conclusion, although the approach has merit and the preliminary experimental results show some promise, the missing baselines, limited novelty, lack of clarity and limited evaluation mean this contribution is not at an appropriate level for such a prestigious conference as ICLR.






**Unimportant typos/comments**:


Inside the caption in Figure 4: “enconder” should be “encoder”

Line 317, "They have gotten good results..." is too informal

The resolution of Figure 3 could be improved.

Line 288: “…major contribution in in image reconstruction…..” Delete one “in”

Line 119: There is a missing space after the period in “on a UoS method.In this manner…”

Line 131: there is a full sentence as follows: “Architectural novelty.” Perhaps the authors meant for this to be a subtitle in boldface?


Line 120: “we can concurrently do both goals”, replace “do” by “perform”

Missing article in the title in line 268


line 281: "there have been numerous research performed...."

**Questions:**

1. Could you explain, clearly and with equations, what your pseudo-completion layer does?

---

### Official Review · Reviewer_MQxx · 2024-10-31

**Soundness:** 2
**Presentation:** 2
**Contribution:** 2
**Rating:** 3
**Confidence:** 3

**Summary:**

This paper studies the problem high-rank matrix completion and extends the union of subspace model with a deep latent model. The union of subspace is done on a non-linear embedding of the original data. To foster the learning of the embedding through auto-encoder, the authors propose a pseudo-completion layer to impute missing entries before feeding into an auto-encoder. The union of subspace constraint is enforced on the latent layer of the auto-encoder. The improved performance is demonstrated on several simulated and real-world datasets.

**Strengths:**

The idea is simple and easy to follow. The proposed algorithm is motivated adequately.

**Weaknesses:**

There is a lack of theoretical investigation of the proposed algorithm. Therefore, the soundness/applicability is questionable. For example, using existing entries in each row to impute the missings in the pseudo-completion layer might cause severe observational bias. More discussions are needed to understand the pros and cons of such an approach. The choices of regularizations and training details (such as initializations, step size, and when to stop) need to be justified either theoretically or empirically. Those weaknesses limit the significance of this work.

**Questions:**

If I understand correctly, $\theta$ is a matrix, what $\ell_1$ and $\ell_2$ norms refer to needs to be clarified.

The writing could be improved, there are many minor errors such as
* Line 92, (Kazdaghli 2023) is not properly cited.
* Line 119, space is missing before "In this manner..."
* Line 131, "Architecture novelty" is not a sentence.
* Figure 4, "Pseudo - completion layer"

---

### Official Review · Reviewer_PZdG · 2024-11-03

**Soundness:** 2
**Presentation:** 2
**Contribution:** 2
**Rating:** 3
**Confidence:** 4

**Summary:**

The paper proposes LUOS model along with its DELUCA neural architecture for matrix completion. The proposed network consisting of pseudo-completion layer, self-expressive layer, and autoencoder architecture is empirically robust to large-scale matrices with high missing ratio. By leveraging the Union of Subspaces in latent space, LUOS addresses complex real-world patterns and improves completion in cases where traditional HRMC methods might struggle.

**Strengths:**

+ Unlike standard linear subspace models, LUOS leverages a non-linear latent space to fit complex structures via deep learning methods, which appears effective in capturing non-linear relationships in incomplete data.

+ LUOS demonstrates substantial improvements over competing models, achieving 40% higher clustering accuracy than the nearest baseline across multiple datasets. Notably, the model showcases strong reconstruction capabilities, even with high levels of missing data.

**Weaknesses:**

- While the latent space transformation in UoS has been claimed to offer flexibility, there lacks theoretical analysis nor empirical validation on how this transformation impacts performance across various types of data distributions.

- The model relies on self-expressive layers, which require access to the full dataset for similarity calculations. This could hinder scalability for extremely large datasets that are not easily fetched into the memory at once.

- The comparative study is not complete. Although the paper includes various imputation and KNN baselines, most of them are too old to represent the state of the art. At least some comparison with traditional (and deep) matrix factorization methods like PMF, RPCA, and NCF should be included.

- The title is not appropriate. It is not informative and does not represent the contribution of the paper.

**Questions:**

I reproduce my questions from the weaknesses:

- While the latent space transformation in UoS has been claimed to offer flexibility, there lacks theoretical analysis nor empirical validation on how this transformation impacts performance across various types of data distributions.

- The model relies on self-expressive layers, which require access to the full dataset for similarity calculations. This could hinder scalability for extremely large datasets that are not easily fetched into the memory at once.

- The comparative study is not complete. Although the paper includes various imputation and KNN baselines, most of them are too old to represent the state of the art. At least some comparison with traditional (and deep) matrix factorization methods like PMF, RPCA, and NCF should be included.

**Details Of Ethics Concerns:**

N/A: it is a basic research.

---

### Meta-Review · Area_Chair_aWkQ · 2024-12-17

**Metareview:**

The paper introduces the Latent Union of Subspaces (LUOS) model for high-rank matrix completion (HRMC). The method is based on combining a pseudo-completion layer that imputes missing data, an auto encoder for learning the latent representation, and a self-expressive layer.

Reviewers generally agreed that the method demonstrated strong empirical performance. However, there are many concerns shared across reviewers on lack of theory, weak baseline, limited novelty, small scale experiment. There were no response from the authors.

**Additional Comments On Reviewer Discussion:**

There was no rebuttal.

---

### Decision · Program_Chairs · 2025-01-22

Reject